# Combination of Plasma-Based Metabolomics and Machine Learning Algorithm Provides a Novel Diagnostic Strategy for Malignant Mesothelioma

**DOI:** 10.3390/diagnostics11071281

**Published:** 2021-07-16

**Authors:** Na Li, Chenxi Yang, Sicheng Zhou, Siyu Song, Yuyao Jin, Ding Wang, Junping Liu, Yun Gao, Haining Yang, Weimin Mao, Zhongjian Chen

**Affiliations:** 1Zhejiang Cancer Research Institute, The Cancer Hospital of the University of Chinese Academy of Sciences (Zhejiang Cancer Hospital), Hangzhou 310022, China; 13777883438@163.com (N.L.); chenxiyang729@gmail.com (C.Y.); seasonjoe1996@gmail.com (S.Z.); sisongucsd@163.com (S.S.); jyyjyy1997@163.com (Y.J.); brcms96@163.com (D.W.); liujunpingzjcc@163.com (J.L.); gaoyun@zjcc.com (Y.G.); 2Institute of Cancer and Basic Medicine (IBMC), Chinese Academy of Sciences, Hangzhou 310000, China; 3The Second Clinical Medical College, Zhejiang Chinese Medical University, Hangzhou 310053, China; 4Department of Pharmaceutical Sciences, Hangzhou Medical College, Hangzhou 310013, China; 5Thoracic Oncology, University of Hawaii Cancer Center, Honolulu, HI 96813, USA

**Keywords:** malignant mesothelioma, metabolomics, machine learning, diagnosis

## Abstract

Background: Malignant mesothelioma (MM) is an aggressive and incurable carcinoma that is primarily caused by asbestos exposure. However, the current diagnostic tool for MM is still under-developed. Therefore, the aim of this study is to explore the diagnostic significance of a strategy that combined plasma-based metabolomics with machine learning algorithms for MM. Methods: Plasma samples collected from 25 MM patients and 32 healthy controls (HCs) were randomly divided into train set and test set, after which analyzation was performed by liquid chromatography-mass spectrometry-based metabolomics. Differential metabolites were screened out from the samples of the train set. Subsequently, metabolite-based diagnostic models, including receiver operating characteristic (ROC) curves and Random Forest model (RF), were established, and their prediction accuracies were calculated for the test set samples. Results: Twenty differential plasma metabolites were annotated in the train set; 10 of these metabolites were validated in the test set. The seven metabolites with most significant diagnostic values were taurocholic acid (accuracy = 0.6429), uracil (accuracy = 0.7143), biliverdin (accuracy = 0.7143), tauroursodeoxycholic acid (accuracy = 0.5000), histidine (accuracy = 0.8571), pyrroline hydroxycarboxylic acid (accuracy = 0.8571), and phenylalanine (accuracy = 0.7857). Furthermore, RF based on 20 annotated metabolites showed a prediction accuracy of 0.9286, and its optimized version achieved 1.0000 in the test set. Moreover, the comparison between the samples of peritoneal MM (*n* = 8) and pleural MM (*n* = 17) illustrated a significant increase in levels of taurocholic acid and tauroursodeoxycholic acid, as well as an evident decrease in biliverdin. Conclusions: Our results revealed the potential diagnostic value of plasma-based metabolomics combined with machine learning for MM. Further research with large sample size is worthy conducting. Moreover, our data demonstrated dysregulated metabolism pathways in MM, which aids in better understanding of molecular mechanisms related to the initiation and development of MM.

## 1. Introduction

Malignant mesothelioma (MM) is an aggressive and incurable malignancy that develops on the lining of the pleural and peritoneal cavities [1,2]. Incidence of MM is strongly associated with exposure to asbestos—a known human carcinogen. Despite the widespread prohibition on the application of asbestos, the worldwide incidence rate of MM is still increasing dramatically as a result of continuous use of asbestos in developing countries and the large existing population with preceding exposure to asbestos [3]. China, as a country with the highest consumption of asbestos, showed a 2.5% increase in incidence rate per year between 2000 and 2013 [4]. Unfortunately, such growth in MM incidence rates is expected to persist in the following decades in China due to its continuous industrial use of asbestos.

The standard treatment for MM involves a combination of surgery, chemotherapy, and radiotherapy. However, efficacy of this multidisciplinary treatment is relatively limited. Recently, novel therapies, including molecular-targeted drugs and immune checkpoint inhibitors, have been applied into practice for treating MM, which, however, turned out to be dissatisfactory [5,6,7]. Nonetheless, MM patients diagnosed in early stages have a higher chance of better prognosis and, accordingly, better survival prospects, illustrating that early detection of MM is extremely important. However, early and accurate diagnosis of MM remains problematic. As suggested in a recent review, the misdiagnosis rate is relatively high worldwide, ranging from 14% in developed countries to 50% in developing countries, reflecting an urgent need for a novel and accurate means of diagnosis [3]. Furthermore, most diagnoses of mesothelioma occur in their advanced stages due to the long latency period (30–50 years after asbestos exposure) and nonspecific symptoms, consequently resulting in delayed treatments and shortened survival time [8]. Thus, it is crucial to identify sensitive and specific biomarkers for the early diagnosis of MM.

Metabolic reprogramming as one of the hallmarks of cancer cells supports the energy formation for their uncontrollable proliferation [9]. Characterization of cancer cells’ metabolic profile not only provides metabolic biomarkers for diagnosis or prognosis, but also reveals the molecular biology of MM that can facilitate the detection of the underlying therapeutic targets [10]. As a powerful tool for the identification and quantification of endogenous metabolites, metabolomics has been used widely in cancer research fields [11,12]. Meanwhile, a machine learning approach has been increasingly utilized in medical field for its value in disease diagnosis. Recently, emerging studies demonstrated that the combination of metabolomics and machine learning is an accurate and effective diagnostic approach for diseases, including cancer [13,14]. Therefore, further exploration on applicability and effectiveness of this combined method for MM is worthwhile.

In this study, a total of 25 malignant mesothelioma patients (MMs) and 32 healthy controls (HCs) were randomly divided into a train set (*n* = 43) and a test set (*n* = 14). An LC-MS-based untargeted metabolomics was performed on the plasma samples. Diagnostic models, including ROC analyses and machine learning algorithm, were established based on the annotated differential metabolites in the train set and further validation of the predictive performance in the test set. Our study not only provides a novel diagnostic strategy for MM, but also reveals the plasma metabolic profile of MM for the first time, providing insights and general evidence for further investigation of molecular biology in MM. However, further studies with a larger sample size on this novel diagnostic method of MM, as well as the panel of dysregulated metabolites, need to be conducted.

## 2. Materials and Methods

### 2.1. Chemicals and Reagents

Acetonitrile (high-performance liquid chromatography (HPLC) grade) and methanol (HPLC grade) were purchased from Tedia Company (Fairfield, OH, USA). Formic acid (HPLC grade) was obtained from Roe Scientific Inc. (Newark, DE, USA). Distilled water was obtained from Wahaha Group Co., Ltd. (Hangzhou, China).

### 2.2. Study Population and Sample Collection

Plasma samples were collected from 25 patients with histopathologic confirmation of MM at Zhejiang Cancer Hospital, China between March 2016 and March 2020, and none had previously received anti-cancer treatment. Plasma controls were collected from 32 age- and sex-matched healthy individuals. All participants were overnight fasted before sample collection. Plasma samples were then immediately centrifuged at 3000 rpm at 4 °C for 15 min. Di-potassium salt of ethylenediaminetetraacetic acid (K2-EDTA) was used as the anticoagulant. Plasma samples were stored at −80 °C until analysis. The detailed characteristics of the participants are listed in Table 1. All participants in this study were provided with informed consent, and the study was conducted in accordance with the ethical standards of the Ethics Committee of Zhejiang Cancer Hospital and the 1964 Helsinki Declaration and its later amendments or comparable ethical standards.

The participants (*n* = 57, MM: 25, HC: 32) were randomly divided into a train set (*n* = 43, MM: 19, HC: 24) and a test set (*n* = 14, MM: 6, HC: 8) using function “*sample*” in R (v3.4.1) (Figure 1).

### 2.3. Sample Preparation

Plasma preparation was performed according to our previous established method [15]. In brief, 320 μL of pre-chilled acetonitrile was added to 80 μL of each plasma sample and vortexed for 30 s. The mixture was centrifuged at 13,000 rpm, 4 °C for 15 min. Then, 350 μL of supernatant was transferred to a new centrifuge tube and lyophilized. The residue was reconstituted with 80 μL of solution consisting of acetonitrile/water (1:4, *v*/*v*). After vortexing for 30 s and centrifuging at 13,000 rpm, 4 °C for 15 min, 60 μL of supernatant was transferred into a vial and 5 μL of the supernatant was analyzed by LC-MS.

Meanwhile, pooled quality control (QC) samples were prepared by pooling equal volume of aliquots from each MM and HC sample followed by extraction as described above. QC samples were analyzed periodically throughout the analytical run to ensure the stability of the platform.

### 2.4. LC-MS Analysis

LC-MS analysis was conducted according to our previous method [16]. Briefly, metabolomic analyses were performed with a Q Exactive^TM^ Hybrid Quadrupole-Orbitrap Mass Spectrometer coupled with an Ultimate 3000 UHPLC system (Thermo Fisher Scientific, CA, USA). An ACQUITY UPLC HSS T3 column (2.1 mm × 100 mm, particle size 1.8 μm, Waters, USA) at 40 °C was used with a flow rate of 0.3 mL/min for chromatographic separation. The mobile phase consisted of 0.1% formic acid (phase A) and acetonitrile (phase B), and the gradient conditions were set as follows: 0–1 min, 2% phase B; 1–10 min, 2–100% phase B; 10–13 min, 100% phase B; 13–13.1 min, 100–2% phase B; 13.1–16 min, 2% phase B. The capillary voltages used for the positive electrospray ionization (ESI+) and negative electrospray ionization (ESI-) modes were 3500 v and 2500 v, respectively. The ion transfer tube temperature was set to 350 °C. The mass scan range (m/z) was set to 70–1000 with a mass resolution of 70,000 in both ESI+ and ESI− modes. The sheath gas was set to 35 and 40 Arb in ESI+ and ESI− modes, respectively. For collecting MS/MS spectra, data-dependent acquisition was performed in a top 10 mode with a mass resolution of 17,500 and stepped collision energies of 10, 20, and 40 eV.

### 2.5. Metabolomic Data Analysis

Mass spectrometry raw data were converted to mzXML format with MSConvert software (http://proteowizard.sourceforge.net/downloads.shhtml, accessed on 15 December 2020), and then analyzed with the R package *XCMS* (v3.4.1) for peak picking, retention time alignment and peak matching. Subsequently, the data were further processed with R package *MetaX* (v1.4.16) for ion filtration with the following criteria: (1) ions that were not detected in over 50% of all QC samples, (2) ions that were not detected in over 80% of all non-QC samples, or (3) ions with a relative standard deviation >30% in QC samples. In order to reduce the influence by the signal drift, quality control-based robust LOESS signal correction algorithm was applied.

Principal component analysis (PCA) and orthogonal partial least squares-discriminant analysis (OPLS-DA) were performed using R package *ropls* (v1.18.8). The differential metabolic features were selected in train set samples with criteria of variable importance in projection values (VIP) > 1 in the OPLS-DA, *p*-value < 0.05 by two-tailed Student’s *t*-test, and fold change >1.5 or <0.667. Metabolites were further annotated by matching their precursor m/z values and MS/MS spectra with reference spectra from online databases METLIN (http://metlin.scripps.edu/, accessed on 10 January 2021) and HMDB (http://www.hmdb.ca/, accessed on 10 January 2021), as well as our in-house database. Metabolism pathway analysis was conducted using the online Metaboanalyst (https://www.metaboanalyst.ca/MetaboAnalyst/home.xhtml, accessed on 8 February 2021).

### 2.6. Metabolite-Based Diagnostic Modeling

For the single metabolite-based models, ROC curve was generated for each metabolite using R package *ROC* (v1.16.2) to evaluate their diagnostic performance in the train set. The top 7 metabolites in terms of AUC values were used for further modeling. Optimal cutoff values of each metabolite were selected through calculating the maximum of Youden’s index (J = sensitivity + specificity − 1). Based on the cutoff values, the unknown samples in the test set were sorted into predictive classes, followed by the calculation of diagnostic performances including accuracy, sensitivity, and specificity.

For the multiple metabolite-based machine learning model, “scaling” and “centering” preprocesses were performed in the train set, with identical preprocessing methods as well as same parameters applied to the test set. Random Forest (RF) algorithm was investigated for the classification of MM and HC via R package *caret* (version 6.0–85). To train the RF model, ten repeated five-fold cross validations were performed along with the automatic optimization of the tunned parameter “mtry”. In order to avoid multicollinearity and complexity of RF model, different variable sets were utilized for prediction. The variable sets dynamically included from top2 to top20 metabolites, which were ranked according to variable importance defined by the RF model.

### 2.7. Statistical Analyses

Statistical analyses were performed using R software (version 3.6.2). ROC analysis was performed by R package *pROC* (version 1.15.3). The Student’s *t*-test was used to compare the means between two groups. A two-tailed *p*-value < 0.05 was considered to be statistically significant.

## 3. Results

### 3.1. Population Characteristics

As shown in Table 1, there were no statistically significant inter-group differences in terms of age and sex (*p*-value > 0.05). Among the MM patients, 17 had pleural MM, and 8 had peritoneal MM. When categorizing by asbestos exposure, 11 reported positive while 10 reported negative, with the other 4 unknown. The total number of asbestos-exposed patients with plural MM was 7, and that of non-exposed ones was also 7. With regard to the 8 patients with peritoneal MM, 4 had been exposed to asbestos, while 3 had not.

### 3.2. Plasma Metabolic Shift between MM and HC Groups

The total ion chromatographs of 5 pooled QC samples in ESI+ mode shown in Figure 2A were indicative of a successfully established LC-MS modality for untargeted metabolomics, where most of the main peaks were consistent with each other in terms of retention time and peak intensity. A total of 4327 metabolic features in ESI+ mode and 6023 in ESI- mode were extracted from the raw data. The clustering of pooled QCs in PCA analysis showed a high degree of consistency which pointed out our results being reliable and reproducible (Figure 2B).

OPLS-DA analysis of the samples of the train set exhibited a significant separation between MM and HC groups, with R^2^ value of 0.78, and cross-validated by 200-time permutation tests (Figure 2C,D). In line, a volcano plot based on *p*-values, VIP values, and fold changes suggested a panel of dysregulated metabolic features in MM plasma samples, with 309 upregulated and 596 downregulated (Figure 2E). A total of 20 metabolites were annotated, with 8 upregulated and 12 downregulated (Table 2).

Validation in the test set further demonstrated that 17 metabolites were of the same variation trend, among which 10 were significant and 7 were not. While an opposite variation trend was observed in test set for tauroursodeoxycholic acid, taurocholic acid, and creatine. Detailed information is listed in Table 2. Moreover, heatmaps in terms of the 20 annotated differential metabolites were plotted for both train and test splits (Figure 3A,B). Both of the plots showed similar patterns and clear distinctions between MM and HC samples. Figure 4A showed a panel of up or downregulated feature metabolites in MM compared to HC. Variation trends were consistent in train and test set for most metabolites, except for tauroursodeoxycholic acid, tauroucholic acid, and biliverdin. Moreover, 14 pathways were enriched for these metabolites, of which most were related to amino acid metabolism and three pathways were significantly enriched, including tryptophan metabolism (*p*-value = 0.0130); beta-alanine metabolism (*p*-value = 0.0269); and phenylalanine, tyrosine and tryptophan biosynthesis (*p*-value = 0.0491) Figure 4B.

### 3.3. Predictive Performance of Single-Metabolite-Based Models and Multiple Metabolite-Based Machine Learning Model

ROC curve analyses were performed with the annotated differential metabolites of the train dataset on a singular basis, and finally seven metabolites were selected being of high AUC values: taurocholic acid (AUC = 0.8421), uracil (AUC = 0.8399), biliverdin (AUC = 0.8289), histidine (AUC = 0.8180), tauroursodeoxycholic acid (AUC = 0.8048), pyrroline hydroxycarboxylic acid (AUC = 0.8026), and phenylalanine (AUC = 0.8004) (Figure 5A–G). Sensitivity and specificity of each metabolite at the optimal cutoff value were provided in Appendix A. Boxplots of the seven metabolites demonstrated a significant difference in circulating levels between MM and HC groups (Figure 5A–G). Specifically, the MM group experienced an upregulation of taurocholic acid, tauroursodeoxycholic acid, pyrroline hydroxycarboxylic acid, and phenylalanine; and a downregulation of uracil, biliverdin, and histidine.

Prediction accuracies were calculated from the test set by applying Youden’s index, showing that some models exhibited low accuracies, which were taurocholic acid (accuracy = 0.6429), uracil (accuracy = 0.7143), biliverdin (accuracy = 0.7143), and tauroursodeoxycholic acid (accuracy = 0.5000), while others still retained relatively high prediction accuracies of over 0.7500. Outstandingly, models of histidine and pyrroline hydroxycarboxylic acid, both reported an accuracy of 0.8571 (Figure 6A).

In addition to ROC models based on single metabolites, the RF model based on all the twenty annotated metabolites was also performed, with a reported AUC_ROC_ value of 1.0000 (Figure 5H). For this model, the optimized value of “mtry” was 2, the prediction accuracy for the train set was 1.0000, and that for the test set was 0.9286 (Figure 6A).

### 3.4. Feature Metabolite Selection for RF Model

Figure 6B–D presented the ranking of the annotated 20 metabolites in RF model, as well as prediction accuracies and kappa values of all models with different amounts of top metabolites. Notably, a model which included the least number of metabolites (i.e., top 2) exhibited a satisfactory prediction accuracy of 0.7857 (Figure 6C,D). For the rest of the models, accuracy improved as more metabolites were included until it reached a maximum of 1.0000 for the model of top 9. Such high accuracy remained from top 10 to top 13 models, after a slight reduction in the model of top 14, the accuracy increased to 1.0000 again for models of top 15 and top 16, and finally dropped to 0.9286 and remained constant for the rest four models (Figure 6C,D). To sum up, reducing metabolites included in RF does not always lower prediction accuracy, and a maximum accuracy of 1.0000 can be reached by inclusion of only 9 metabolites.

### 3.5. Differential Metabolites between Peritoneal and Pleural MMs

When comparing the levels of 20 differential metabolites between peritoneal (*n* = 8) and pleural (*n* = 17) MMs, tauroursodeoxycholic acid, tauroucholic acid, and biliverdin were found to be significantly upregulated in peritoneal MMs (Figure 7A–C). Comparison between overall MMs (*n* = 25) and HCs (*n* = 32) revealed that the levels of tauroursodeoxycholic acid and tauroucholic acid were both upregulated in MMs, in line with the variation trend observed in the train set (Figure 5A,E and Figure 7A,B), while the concentration of biliverdin was downregulated in MMs, in consistent with the change in the train set (Figure 5C and Figure 7C). Further analyses of the difference between the two subtypes of MMs and HCs demonstrated that the levels of tauroursodeoxycholic acid and tauroucholic acid in peritoneal MMs were significantly higher than HCs, while those in pleural MMs were not (Figure 7A,B). The levels of biliverdin in pleural MMs were significantly downregulated compared to HCs, while those in peritoneal MMs were not (Figure 7C).

## 4. Discussion

Mesothelioma is rare cancer that mainly results from exposure to asbestos. Despite asbestos being banned in most countries, asbestos-induced mesothelioma still cannot be depleted. Apart from the long incubation period, mesothelioma is distinctive of its difficulties in diagnosis and treatment, hence leading to an unfavored late-stage diagnosis and poor overall survival. It was proved that early diagnosis significantly improves the overall survival of MM [3]. However, the currently available diagnostic methods are reported as being of relatively poor sensitivity and specificity [17].

Herein, aiming to improve MM prognosis by optimizing clinical diagnosis, this study used a combined strategy of untargeted metabolomics and machine learning algorithms to investigate differential metabolites between plasma samples of MM patients and age- and sex-matched HCs. After screening and annotation, twenty metabolites were found to be dysregulated, of which ten metabolites were validated of their variation trends in the test set. And to better elucidate the diagnostic significance of these metabolites, both single metabolite-based ROC model and multiple metabolite-based RF discriminated between MM and HC, and the RF exhibited higher prediction accuracy than the single metabolite-based ROC model, hence we assumed that: (a) Machine learning models combined with metabolomics is a novel and promising measure for accurate mesothelioma diagnosis; (b) our annotated metabolites are of high diagnostic value clinically, as well as giving instructions and supportive evidence for future research.

Biliverdin and bilirubin are two feature metabolites of mesothelioma that have potential diagnostic value, supported by the result that biliverdin ranked fourth in the RF model in terms of overall importance. Biliverdin is the precursor of bilirubin, which is initially released during catabolism of red blood cells and then reduced to bilirubin, in line, our results suggested the same variation trend in bilirubin and biliverdin [18]. Bilirubin is widely known for its beneficial effects on health as an antioxidant in that a slightly higher level helps prevent various morbidities, including cancer [19,20]. In contrast, a low plasma level of bilirubin could be a risk factor for various cancers, such as colon cancer [20] and lung cancer [21], etc. Consistently, downregulations in biliverdin and bilirubin in MM patients revealed in this study suggested a possibility of these metabolites being potential biomarkers of malignant mesothelioma. Furthermore, exposure to asbestos, which is the major culprit of mesothelioma, can lead to adsorption of hemoglobin and histones to the inhaled asbestos fibers, causing oxidative stress and finally cancer. This might explain the observed reduction in plasma bilirubin and biliverdin levels in mesothelioma patients compared to healthy participants. However, future research is needed to validate this hypothesis and to investigate further mechanisms of bilirubin’s protective effect against malignancy on a molecular basis.

Intemperate utilization of amino acids is a phenomenon well recognized and validated in many cancers [22,23]. Tryptophan is an essential amino acid taken up exclusively from diet and serves as a building block for protein biosynthesis [24]. It is mainly involved in two metabolic pathways, the serotonin pathway, and the kynurenine pathway [25], of which the latter is noticed to be deregulated in various malignancies as a result of aberrant activation of IDO that subsequently leads to immunosuppression [24]. In line, our results detected enrichment of this pathway, as well as a significantly increased circulating level of kynurenine in MM patients by 1.68 folds in the train set and 1.9 folds in the test set. The breakdown product of the serotonin pathway, 5-hydroxyindoleacetic acid, was also detected to be upregulated in the present study, thus further reflecting a phenomenon that tryptophan was utilized to a greater extent in MM compared to HC. Meanwhile, a semi-essential amino acid, histidine, had reduced 0.64 folds in the train set and 0.46 folds in the test set, which might imply a favored uptake and utilization of this amino acid by cancer cells. Tumor’s glutamine dependency may explicate its appetite for histidine, as the catabolism of this amino acid provides free ammonia essential for glutamine biosynthesis [26]. Moreover, upregulation of pyrroline hydroxycarboxylic acid revealed an upregulated proline consumption by the cancer cell, because pyrroline hydroxycarboxylic acid is generated by oxidation of pyrroline-carboxylate, which is an intermediate product of proline metabolism [27]. An increase in phenylalanine level was observed in MM, which was contradictory to variation in histidine. Thereby, these metabolites, and their associated amino acids, serve as potential diagnostic biomarkers to aid clinical distinction between HC and MM, though further research is still required to reveal underlying mechanisms.

Taurocholic acid and tauroursodeoxycholic acid are bile acids that exist usually as sodium salts of bile. Physiological functions with regard to tauroursodeoxycholic acid have remained unclear in MM, while it was suggested a protective effect of this acid against bile acid-induced apoptosis [28], which supported the initiation of colon cancer [29]. In addition, epidemiological evidence showed that taurocholic acid is associated with tumorigenesis through the generation of hydrogen sulfide, which is a genotoxic compound and tumor-promotor [30]. In the present study taurocholic acid and tauroursodeoxycholic acid were found significantly upregulated in MM, suggesting their potential of being parameters for monitoring cancer risk and prognosis. Nevertheless, inconsistency in the variation trend of these metabolites was observed between samples from the train set and samples from the test set. Such phenomenon was probably due to an uneven distribution of peritoneal MM patients (who had much higher plasma levels of bile acids than plural MM and HCs) allocated to the train set and test set: only one peritoneal MM patients were included in the test set, comparing to 7 included in the train set, thus lowering the average levels of these acids in MM of the test set. Altogether, taurocholic acid and tauroursodeoxycholic acid are potential biomarkers to discriminate between MM and HC, as well as discrimination between subtypes of MM.

Apart from the aforementioned metabolites, plasma metabolomic profiling of this study also demonstrated upregulation in glycocholic acid, and oleamide, as well as downregulation in 3-guanidinopropanoate, dihydroxybenzoic acid, malic acid, androsterone sulfate, dehydroepiandrosterone, prasterone sulfate, uracil, and uridine in MM samples of both train and test sets. While creatine exhibited inconsistent trends, that is downregulated in the train set and upregulated in the test set. Herein the present study demonstrated the potential of these metabolites of being diagnostic biomarkers, though mechanisms have not yet been fully explained and further research is required to investigate them and to validate these trends.

Limitations of this study must be mentioned. First, the sample size of this study is relatively small for machine learning algorithms. Therefore, this study may only be considered a pilot study due to its small sample size. A larger cohort is needed to validate this novel diagnostic strategy for MM. Due to the rarity of MM, a large sample size-based metabolomics might be only achieved through multiple-center cooperation. Second, the predictive performance of RF is limited due to only two classes of participants involved in the machine learning model. To reduce the probability of misdiagnosis, further study should involve more classes of samples, such as asbestos-exposed individuals and cancers that are easily misdiagnosed with MM (lung cancer or ovarian cancer). Third, the amount of metabolite, which was annotated in this study, was relatively low due to a lack of available MS2 spectra, leading to many feature ions with potential diagnostic value not being annotated. Last but not least, the potential biological function and molecular mechanisms of differential circulating metabolites, such as kynurenine, bilirubin, and biliverdin, are required to be clarified further.

In conclusion, this study aimed to investigate the plasma metabolic profile in MM patients for the first time; the data revealed several dysregulated metabolism pathways, including tryptophan metabolism and β-alanine metabolism, as well as phenylalanine, tyrosine, and tryptophan biosynthesis, aiding in understanding the molecular biology of MM and providing potential therapeutic targets for MM. Furthermore, the combination of metabolomics and machine learning exhibited an outstanding diagnostic value for MM, pointing out a novel and effective strategy for MM diagnosis. Future studies with a large sample size are encouraged to start soon on this rare disease, which could significantly benefit the prognosis of MM patients.

## Figures and Tables

**Figure 1 diagnostics-11-01281-f001:**
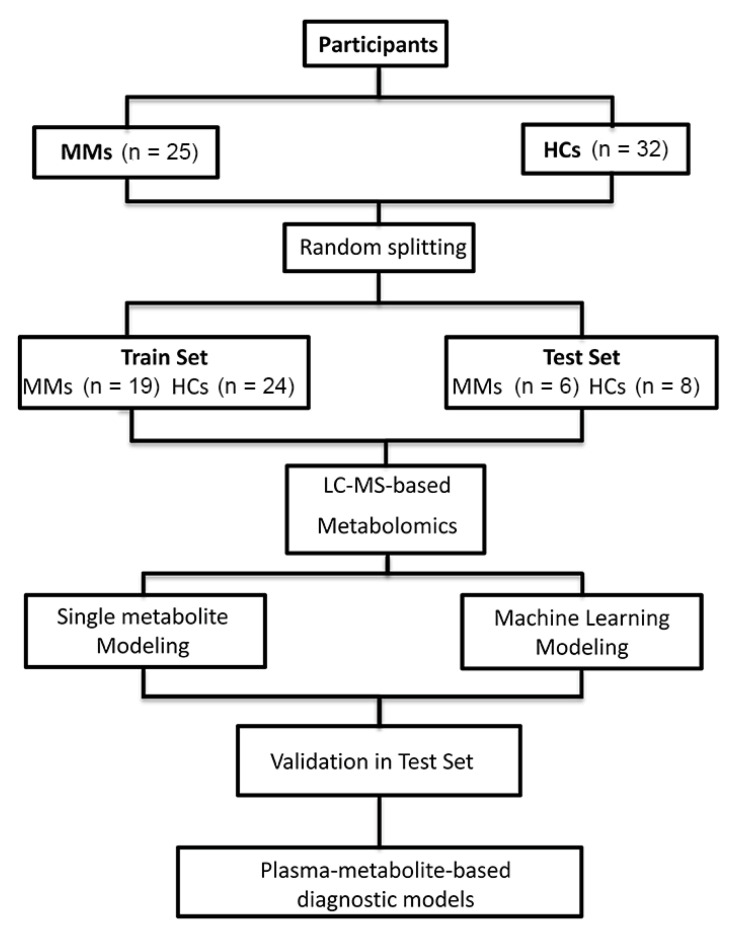
Flowchart of this study.

**Figure 2 diagnostics-11-01281-f002:**
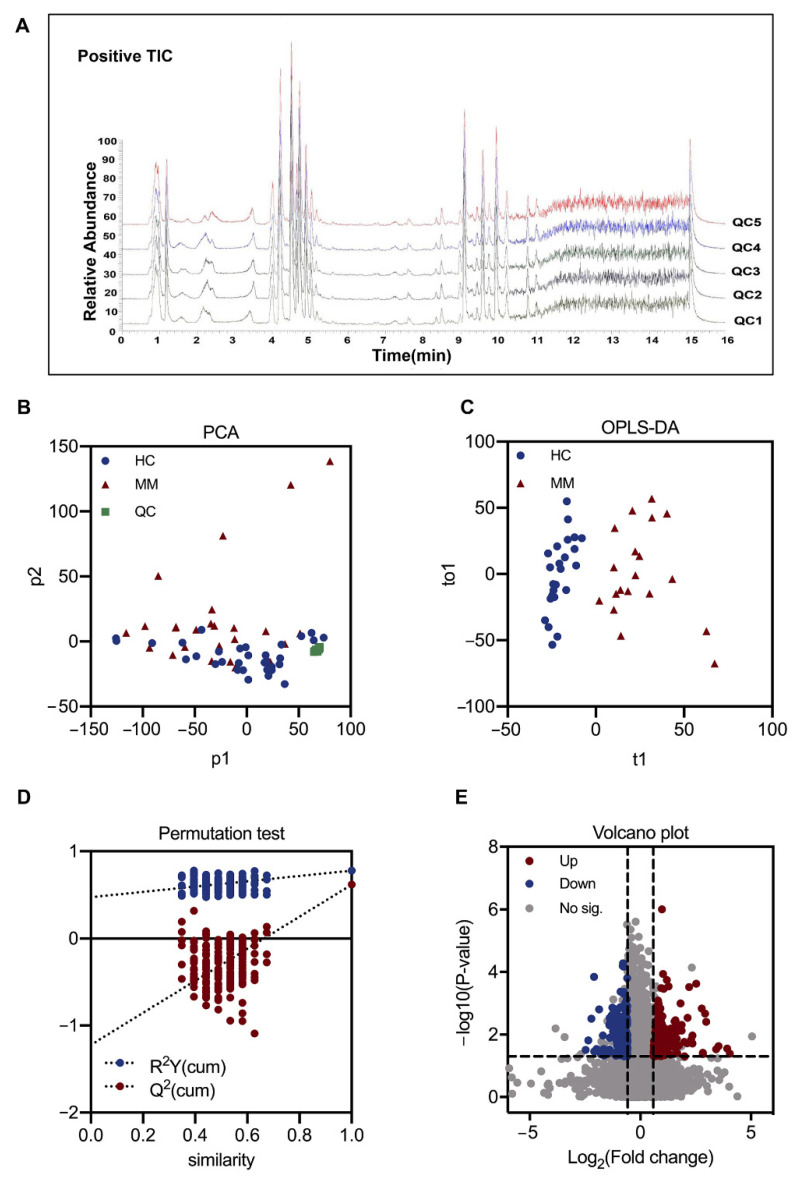
LC-MS analysis and metabolomics analysis for screening out the differential metabolic features between MM and HC groups. (**A**) Typical total ion chromatograms of pooled-QC sample in positive electrospray ionization (ESI+) mode. (**B**) PCA score plot for all the samples, including QC samples. (**C**) OPLS-DA score plot for MM and HC groups. (**D**) Permutation test with 200-time cross-validation for OPLS-DA. (**E**) Volcano plot for the dysregulated ions with criteria of *p*-value < 0.05, VIP > 1.0 and fold change >1.5 or <0.667. PCA: principal component analysis. OPLS-DA: orthogonal partial least squares-discriminant analysis. VIP: variable importance in projection.

**Figure 3 diagnostics-11-01281-f003:**
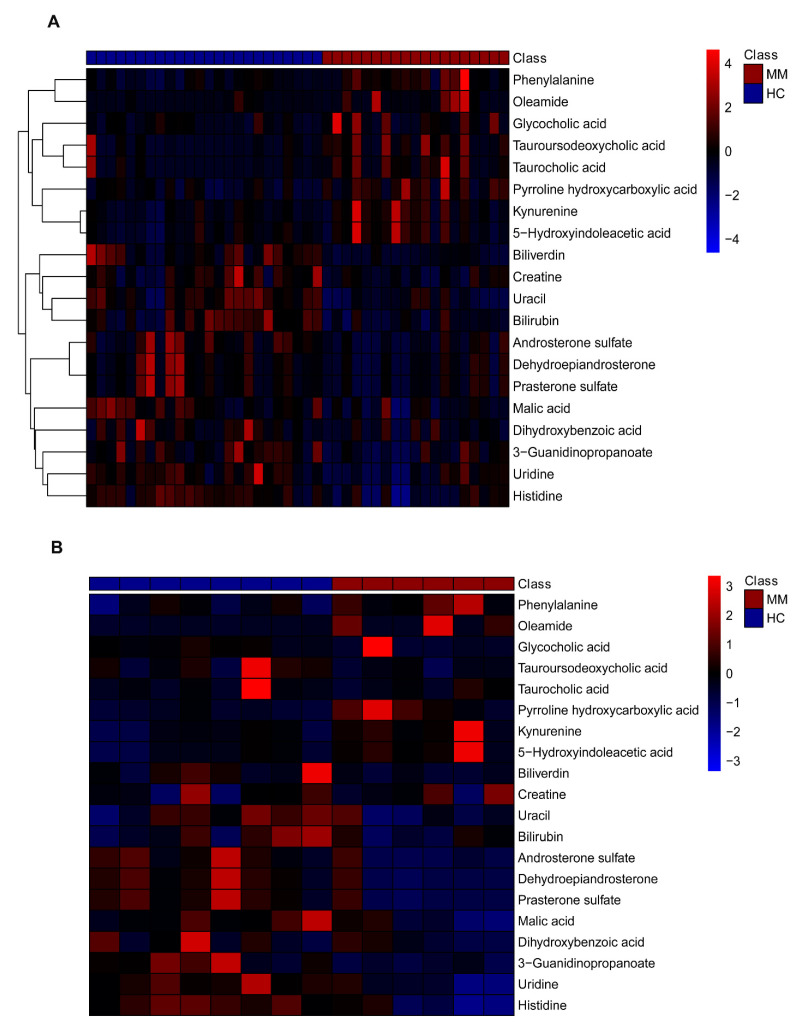
Heatmaps of 20 annotated differential metabolites between MM and HC groups in both train set (**A**) and test set (**B**). Clustering by row (metabolite) was performed only for train set.

**Figure 4 diagnostics-11-01281-f004:**
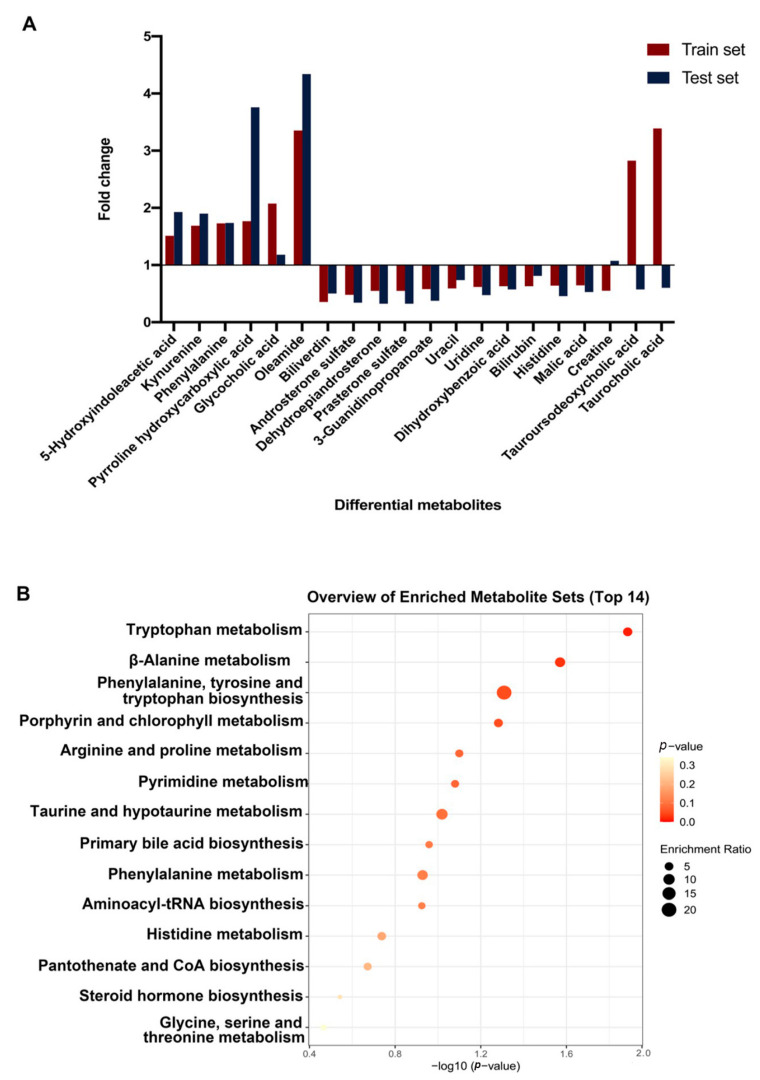
Fold changes of 20 annotated differential metabolites between MM and HC groups in both train set and test set (**A**) and pathway analysis of the 20 metabolites (**B**).

**Figure 5 diagnostics-11-01281-f005:**
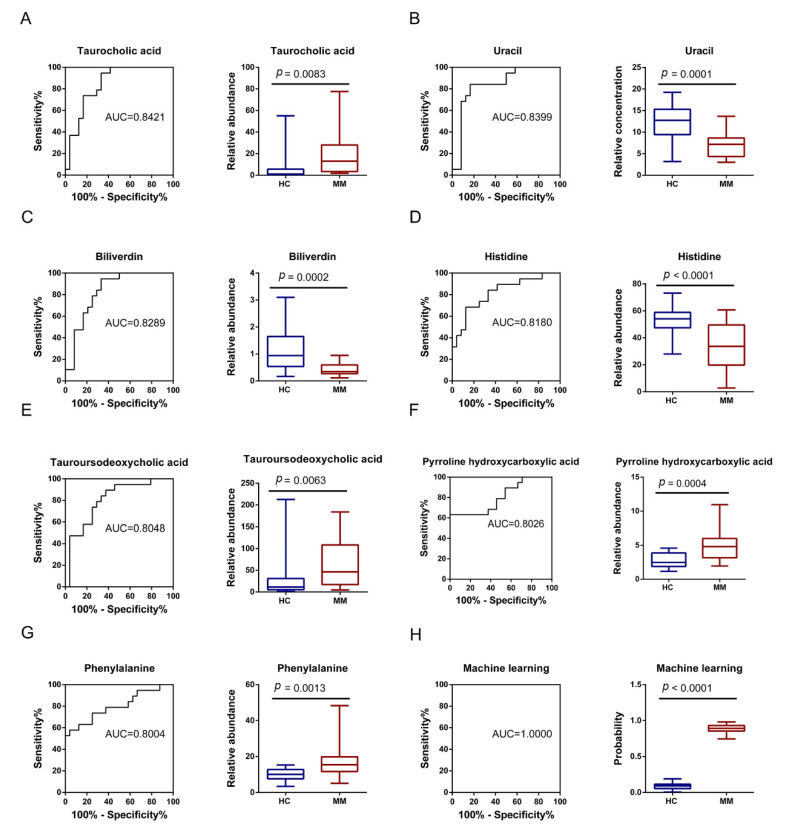
Seven single-metabolite-based receiver operating characteristics (ROC) curves of AUC > 0.8 and multiple metabolite-based machine learning model, and corresponding boxplots according to peak intensity or prediction probabilities in the train set. (**A**) Taurocholic acid. (**B**) Uracil. (**C**) Biliverdin. (**D**) Histidine. (**E**) Tauroursodeoxycholic acid. (**F**) Pyrroline hydroxycarboxylic acid. (**G**) Phenylalanine. (**H**) Machine learning model (Random Forest algorithm).

**Figure 6 diagnostics-11-01281-f006:**
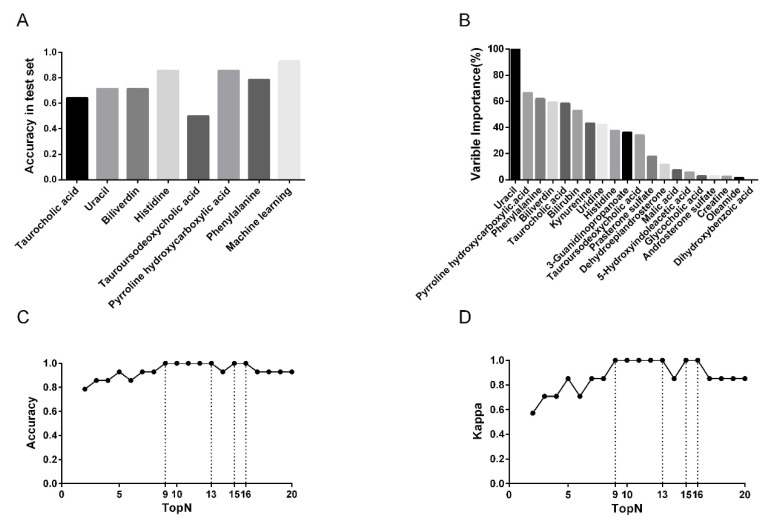
Model comparison and feature selection in machine learning model. (**A**) Comparison of prediction accuracy among single metabolite-based ROC curve models and multiple metabolite-based machine learning model (Random Forest model, RF). (**B**) variable importance of 20 metabolites in RF were ranked. The curves of predictive accuracy (**C**) and Kappa (**D**) values increase as the number of feature metabolites grows.

**Figure 7 diagnostics-11-01281-f007:**
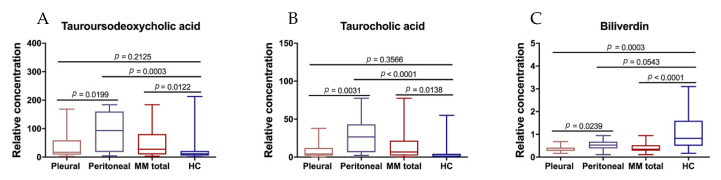
Comparisons of tauroursodeoxycholic acid (**A**), tauroucholic acid (**B**), and biliverdin (**C**) among pleural malignant mesothelioma patients (MMs) (*n* = 17), peritoneal MMs (*n* = 8), total MMs (*n* = 25), and healthy controls (HCs) (*n* = 32). Two-tailed student’s t-test was used to compare the means and *p*-value < 0.05 was recognized as significant.

**Table 1 diagnostics-11-01281-t001:** Clinical characteristics of patients with malignant mesothelioma (MM) and healthy controls (HC).

Feature	HC (*n* = 32)	MM (*n* = 25)	*p*-Value ^a^
Gender			0.83
Male	17 (53.1%)	14 (56.0%)	
Female	15 (46.9%)	11 (44.0%)	
Age			0.99
Mean ± SD	55.8 ± 7.4	55.7 ± 10.9	
Site			NA
Pleural mesothelioma	NA	17 (68.0%)	
Peritoneal mesothelioma	NA	8 (32.0%)	
Asbestos exposure			NA
Yes	NA	11 (44.0%)	
No	NA	10 (40.0%)	
Unknown	NA	4 (16.0%)	

^a^: Two tailed Chi-square test was used to compare the distribution of sex or age between two groups; *p*-value < 0.05 was recognized as significant. MM: malignant mesothelioma. HC: healthy controls. SD: standard deviation. NA: not available.

**Table 2 diagnostics-11-01281-t002:** Twenty annotated differential circulating metabolites between MM patients and healthy controls.

Metabolite	Mode ^a^	m/z ^b^	rt (min) ^c^	Train Set ^d^			Test Set	
				VIP ^e^	*p*-Value	FC ^f^	*p*-Value	FC ^f^
Histidine ^g^	Neg	154.061	0.86	2.06	7.36 × 10^−5^	0.64	2.27 × 10^−3^	0.46
Uracil	Pos	113.035	2.16	1.52	1.21 × 10^−4^	0.59	1.09 × 10^−1^	0.74
Biliverdin	Pos	639.407	11	1.34	2.23 × 10^−4^	0.36	1.33 × 10^−1^	0.5
Pyrroline hydroxycarboxylic acid ^g^	Pos	130.05	5.16	2.05	4.42 × 10^−4^	1.77	1.66 × 10^−2^	3.76
Bilirubin	Pos	585.27	14.42	1.31	6.79 × 10^−4^	0.63	5.03 × 10^−1^	0.81
Phenylalanine ^g^	Pos	166.086	4.57	2.10	1.31 × 10^−3^	1.73	2.72 × 10^−2^	1.74
Uridine ^g^	Neg	243.063	2.12	1.06	2.87 × 10^−3^	0.62	1.27 × 10^−2^	0.48
Kynurenine	Pos	209.092	3.96	1.38	3.46 × 10^−3^	1.69	5.67 × 10^−2^	1.9
Malic acid	Neg	133.013	1.15	1.45	3.66 × 10^−3^	0.65	5.61 × 10^−2^	0.53
Androsterone sulfate ^g^	Neg	369.176	6.72	1.27	5.62 × 10^−3^	0.48	3.66 × 10^−2^	0.34
Tauroursodeoxycholic acid	Neg	498.292	7.53	1.52	6.26 × 10^−3^	2.82	1.63 × 10^−1^	0.58
3-Guanidinopropanoate ^g^	Neg	130.061	0.97	1.14	7.81 × 10^−3^	0.58	2.07 × 10^−2^	0.37
Taurocholic acid	Neg	514.287	6.92	1.82	8.26 × 10^−3^	3.39	5.49 × 10^−1^	0.6
5-Hydroxyindoleacetic acid ^g^	Pos	192.065	3.96	1.16	1.69 × 10^−2^	1.51	4.85 × 10^−2^	1.93
Oleamide ^g^	Pos	282.279	12.16	1.73	2.11 × 10^−2^	3.35	2.25 × 10^−2^	4.34
Glycocholic acid	Neg	464.304	6.71	1.01	2.16 × 10^−2^	2.07	8.21 × 10^−1^	1.18
Dehydroepiandrosterone ^g^	Neg	367.16	7.28	1.12	2.59 × 10^−2^	0.55	1.88 × 10^−2^	0.33
Prasterone sulfate ^g^	Neg	367.16	7.28	1.12	2.59 × 10^−2^	0.55	1.88 × 10^−2^	0.33
Creatine	Pos	132.077	0.97	1.32	2.85 × 10^−2^	0.55	8.72 × 10^−1^	1.07
Dihydroxybenzoic acid	Neg	153.019	5.08	1.12	4.96 × 10^−2^	0.63	3.33 × 10^−1^	0.57

^a^: “Pos” and “Neg” refer to positive scan mode and negative scan mode in mass spectrometry, respectively. ^b^: Mass-to-charge ratio. ^c^: Retention time. ^d^: Differential metabolite features selected based on train set samples by criteria of VIP > 1.00, *p*-value < 0.05, and (FC > 1.5 or FC < 0.667). ^e^: Variable importance in projection (VIP) values from orthogonal partial least squares-discriminant analysis (OPLS-DA). ^f^: Fold change. ^g^: Differential metabolites that were validated in the test set in terms of fold changes and *p*-values.

## Data Availability

The authors confirm that the data supporting the findings of this study are available within the article [and/or] its Appendix A. Raw data were generated at the Cancer Hospital of the University of Chinese Academy of Sciences (Zhejiang Cancer Hospital). Derived data supporting the findings of this study are available from the corresponding authors on request.

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
