# Peer review of "Combination of Plasma-Based Metabolomics and Machine Learning Algorithm Provides a Novel Diagnostic Strategy for Malignant Mesothelioma"

_diagnostics, 2021, doi:10.3390/diagnostics11071281_

Round 1
Reviewer 1 Report
Machine learning based on plasma metabolites is a promising diagnostic approach to cancer diagnosis. This approach was used in the present study to identify relevant biomarkers in the near future to improve the prognosis of MM by optimizing clinical diagnosis.
The present study has some limitations that need to be acknowledged.
- First, it is difficult to draw a firm conclusion due to the small number of patients included in the study, I would call it a pilot study and I would reorganize the discussion better based on this comment.
- Second, the patients in the control group (Healthy Control) are more than the MM patients, it is not clear why?
- In the ROC curves the authors report only the AUC (area under the curve) in the graph, I would suggest adding the percentage of specificity and sensitivity respectively.
In conclusion, by accumulating more reliable data, this novel methodology may also be able to predict the efficacy of each therapy. Therefore, the manuscript may be accepted after fewer recommended revisions.
Author Response
Dear reviewer,
Thank you for your suggestions. Please see below point-by-point responses:
The present study has some limitations that need to be acknowledged.
- First, it is difficult to draw a firm conclusion due to the small number of patients included in the study, I would call it a pilot study and I would reorganize the discussion better based on this comment.
Yes, because MM is so rare and often misdiagnosed, it is indeed difficult to include a large cohort in one study. We had now indicated this in the limitation paragraph in the discussion section of the revised manuscript.
- Second, the patients in the control group (Healthy Control) are more than the MM patients, it is not clear why?
We planned to recruit 32 pairs of samples, however, due to the rarity of this malignancy, we only managed to collect 25 mesothelioma samples. Fortunately, the diseased samples and control samples were matched in age and gender.
- In the ROC curves the authors report only the AUC (area under the curve) in the graph, I would suggest adding the percentage of specificity and sensitivity respectively.
Thank you for the suggestion, we had provided the data in supplementary table.
Please find attached the revised manuscript, with a supplementary table ('Table S1') added on the last page. Also, all changes in the revised manuscript is marked in blue.
Best regards,
Zhongjian Chen
Reviewer 2 Report
The authors present their study regarding a metabolamics based diagnostic model to predict the presence of malignant mesothelioma. The study is well constructed and the the authors have provided careful consideration of their results and have approached their hypothesis in a sound methodological manner. There are no significant concerns to be addressed at this time and the manuscript is acceptable for publication
Author Response
Dear reviewer,
Thank you very much for your comment.
Best regards,
Zhongjian Chen